# Teaching Machine Learning for the Physical Sciences:
# A summary of lessons learned and challenges

**Viviana Acquaviva** [1] [2]

## Abstract

This paper summarizes some challenges encountered and best practices established in several years of teaching Machine Learning for the Physical Sciences at the undergraduate and graduate level. I discuss motivations for teaching ML to physicists, desirable properties of pedagogical materials, such as accessibility, relevance, and likeness to real-world research problems, and give examples of components of teaching units.

## 1. ML x Physical Sciences

Machine learning methods have become ubiquitous in many data-intensive disciplines, including, of course, Physics and Astronomy. The Physical sciences offer a rich landscape of observational and simulated data that are suitable to be analyzed using machine learning and deep learning tools. Identifying particles produced in collision events at the Large Hadron Collider, processing astronomical images from large surveys, identifying transient phenomena in real time, creating fast approximated solutions for lattice theories, or building "emulators" for expensive cosmological simulations are just some of the uses that have become popular in the last decade (see *e.g.* Carleo et al. 2019 for a review).

It follows as a logical consequence that the foundations of machine learning methods should be taught as part of the standard Physics curriculum. In part, this is because they are bound to become standard tools for Physics research. Even more importantly, they are a great pedagogical tool to stimulate critical thinking and to build transferable skills that can create better job perspective for Physics graduates, by leveraging the enormous growth of jobs in the Data Science area that are accessible to those with a rigorous scientific background and strong computational skills.

[1] Physics Department, NYC College of Technology, 300 Jay Street, Brooklyn, NY 11201, USA [2]Center for Computational Astrophysics, Flatiron Institute, New York, NY 10010, USA. Correspondence to: Viviana Acquaviva <vacquaviva@citytech.cuny.edu>.

*Proceedings of the $2^{nd}$ Teaching in Machine Learning Workshop*, PMLR, 2021. Copyright 2021 by the author(s).

## 2. Teaching ML to physicists

There are many great learning resources that either low-cost or free. These include excellent books with free online Jupyter notebooks (Géron, 2019; VanderPlas, 2016), courses on online platforms like Coursera or Udemy, and chances to practice on fairly complex data sets such as those hosted by Kaggle. However, the abundance of choices can be overwhelming for beginner practitioners, who would have to "mix-and-match" different resources to create a curriculum, and more importantly, resources that are tailored to the process of scientific research, and, in particular, to the physical sciences, are still scarce; (Ivezić et al., 2014) is a happy exception, focused on Astronomy. Furthermore, if we want machine learning to become a standard part of the Physics curriculum, we need to provide resources for instructors: many of them won't have been trained in this subject during their own course of study, so that *lowering the barrier for teaching is as important than lowering the barrier for learning*. Five years ago, I started creating materials for a "ML for Physics and Astronomy" course, almost from scratch. Since than, I have taught this class several times, at the undergraduate level to STEM majors, and at the graduate level to Physicists and Astronomers, and I have written the first draft of a textbook on the same subject. Here, I'd like to share some of the practices I have found to be useful and some of the challenges I have identified during this time.

### 2.1. Needs

These are some desirable qualities of materials used to teach Machine Learning to Physics (or more in general, STEM) students:

**Accessibility:** While most STEM majors are familiar with linear algebra, calculus, and statistics, there is great variability in the level of mathematics they are able or willing to handle. At the undergraduate level, I advocate for keeping the complex mathematics at a minimum, and focus on the conceptual aspects of how different algorithms work. My experience is that this approach is the most inclusive and still provides a good foundation to those who would like to explore a topic in higher detail.

**Ok, we are now ready to run our first Random Forest model!**

To get an idea of what we are shooting for, we can look at the figure of the paper.

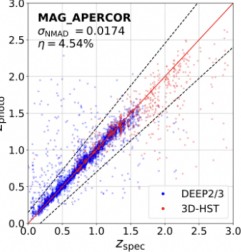

In the figure above, $\sigma_{NMAD}$ is the normalized median absolute deviation of the residual vector, and $\eta$ is the fraction of outliers, defined as those objects for which (z_true - z_est)/(1+z_true) > 0.15.

```
]:  model = RandomForestRegressor()
```

```
]:  model.get_params()
```

**Task (15 mins):**

Establish benchmark score for a RF algorithm with default parameters, using the "cross_validate" function with with 5-fold cross validation strategy. Diagnose whether the trained model suffers from high bias or high variance, and discuss strategies for improvement.

*Figure 1.* An excerpt from a lecture notebook, including the "target performance" from (Zhou et al., 2019), and an example task that students may be asked to complete during class.

**Relevance:** This is possibly the greatest challenge - finding problems and data sets that are relevant to scientific research is hard! Most of the "introductory" data sets (MNIST, Boston housing data, Iris...) are too simple, and working with those does not resemble the typical challenges of a research problem. Many more advanced ones (e.g., those found on Kaggle, which include several Physics/Astronomy challenges) are very complex, are require a vast amount of background information. Striking the right balance of finding data and problems that are beginner-friendly, while at the same time presenting some nontrivial features and help develop intuition, is difficult.

**Real-world likeness:** Experienced machine learning practitioners would often say that "prepping" the data is one of the most time-consuming and important tasks in a ML project. I wholeheartedly agree. Many existing data sets for beginner practitioners are very "curated", and this limits their usefulness in demonstrating best practices in very important aspects such as data exploration, data cleaning, transformations, imputing strategies, and feature engineering. Additionally, themes that are of fundamental importance in research, such as choosing or building an appropriate evaluation metric or estimating uncertainties, are often absent from pedagogical material.

How can we put together materials that have all these desir-

able properties? In short, it takes a lot of work, and it takes a village. As I was assembling resources, I was grateful to count on the help of a very supportive Physics/Astronomy community.

An approach that I have found to be promising is to start with a literature review, and find papers that apply a certain method to data of interest. Some of the most successful pedagogical examples I have used propose to match, or improve on, the performance of a published paper. This is appealing to students because it communicates that they are doing quality work (or at least, that the ML aspect is at a publication-worthy level). For several exercises in my upcoming book, I have reached out to authors of a paper, and asked for access to data, if necessary, and for some introductory guidelines to create a problem-solving pipeline. In class, I show students what our final goal (performance-wise) would be, and try to build the pipeline together with them. I have found that if I propose a solution, it's always helpful to riddle it with poor choices; for example, "forgetting" to normalize the data when needed, including noisy or very correlated features, using accuracy on imbalanced data sets... The process of improving the code helps students retain a stronger memory of these potential pitfalls. Colleagues have indicated that they use a similar approach successfully (S. Caron, private comm.)

*Table 1.* Example assignment for the teaching unit presented in the text.

| DESCRIPTION | TASK |
|---|---|
| START FROM THE FULL DATA SET. WE SAW IN THE LECTURE NOTEBOOK THAT THE PERFORMANCE CHANGES A LOT ONCE THE SELECTION CRITERIA ARE APPLIED. | • FIGURE OUT WHICH OF THE DATA CLEANING CUTS WE MADE WAS THE MOST SIGNIFICANT IN TERMS OF IMPROVING THE SCORES OF THE FINAL MODEL. |
| NOW CHOOSE ONE REFERENCE ALGORITHM AMONG THE ONES WE SAW (RF, ADABOOST, GBM), AND REFER TO THE OPTIMIZED MODEL (NO NEED TO RE-RUN THE GRID SEARCH). LET'S CALL THIS MODEL 1. USE THE DATA SET SELECTION WITH 6,307 OBJECTS AND 6 FEATURES. | • GENERATE PREDICTIONS USING CV AND PLOT THEM IN A HISTOGRAM, TOGETHER WITH THE TRUE VALUES. WHICH DISTRIBUTION IS NARROWER? EXPLAIN WHY.

• OPTIMIZE (USING A GRID SEARCH FOR THE PARAMETERS YOU DEEM TO BE MOST RELEVANT) THE EXTREMELY RANDOM TREE ALGORITHM AND COMPUTE THE OUTLIER FRACTION AND NMAD. HOW DOES IT COMPARE TO MODEL 1? COMMENT NOT JUST ON THE SCORING PARAMETER(S), BUT ALSO ON VARIANCE/BIAS. WHICH ONE WOULD YOU PICK?

• IN THE PAPER THAT WE USED AS A REFERENCE (ZHOU ET AL., 2019), THE AUTHORS ACTUALLY USE COLORS, NOT MAGNITUDES, AS FEATURES. DISCUSS WHY THIS MIGHT BE A BETTER CHOICE.

• FIND IN THE PAPER THE EXACT LIST OF FEATURES, AND GENERATE THEM. ARMED WITH YOUR NEW SET OF FEATURES, USE AN ALGORITHM OF YOUR CHOICE TO MATCH OR BEAT THE PERFORMANCE QUOTED IN THE PAPER (NMAD: 0.0174; OLF, 4.54%).

• DO YOU HAVE ANY IDEAS TO GENERATE FEATURES THAT COULD BE USEFUL? IF SO, FEEL FREE TO GO AHEAD AND REPORT ANY IMPROVEMENT (OR LACK THEREOF). YOU SHOULD MOTIVATE YOUR CHOICE OF FEATURES! |

## 3. Elements of a teaching unit

The strategies I have developed are especially tailored to introductory Machine Learning courses at the advanced undergraduate/early graduate level in "official", grade-bearing classes. I find it helpful to integrate some traditional coursework elements (for example, slides and review questions) with more hands-on content, such as lecture notebooks and programming worksheets assignments. Often, the two sets "talk" to each other; for example, homework might include completing parts of a lecture notebook. For more advanced practitioners or more informal settings, such as summer schools, I usually skip the assignments and try to make each unit self-contained.

My approach is to present each machine learning theme (typically, a new algorithm or a discipline-wide concept, such as cross-validation or hyperparameter optimization) *in parallel with* a Physics or Astronomy problem (examples I have used recently include identifying potentially habitable planets, classifying the products of collision events in particle physics data, or creating a model for the rise of water in different US stations). The example materials shown here are excerpts from a teaching unit where we discuss and use ensemble methods (in particular, bagging and boosting algorithms) to solve the problem of determining distance (parameterized by *redshift, z*) to faraway galaxies from the shape of their spectral energy emission.

The elements of each teaching unit are the following:

**A Power Point (or equivalent) presentation**, with some blackboard discussion, which introduces the ML theme from a theoretical perspective, as well as the data set used;

**A Jupyter notebook lecture**, with some blank parts "to fill". The notebook shows the process of problem-solving - for example, setting up and optimizing the new ML method, or establishing good practices. The extent of the "fillable" parts varies according to the class and the time availability. When possible, I like to do mini assignments (10-15 minutes), where students are asked to complete a single task - for example, do some exploratory data analysis, find a "bug" in my code, or improve on a benchmark performance. An example, including the "target performance" from a published paper (Zhou et al., 2019) and a "mini-task", is shown in Fig. 1.

**Quizzes/review questions**; these are ungraded and usually a students' favorite. Most of them are in multiple-choice form and are meant to reinforce the theoretical foundations, for example with questions on the suitability of a given method to a given problem, or the parameters of a specific method. In a classroom setting, these usually work well as think/pair/share material; I have also organized them as group competitions for extra credit. They also lend themselves to be used in online settings with interactive lecture softwares like Sli.do, Mentimeter, or similar.

**Reading material**; I include traditional (*e.g.* book chapters, journal articles) and less traditional (*e.g.* blog posts, YouTube videos) resources for every unit. Learners have

different learning preferences/styles and I feel that it is important for them to have several options. Some of the sources I recommend for this unit are (Louppe, 2014) and the relevant part of Chapter 5 of (VanderPlas, 2016), but also this YouTube video, this blog post and this one, and this notebook.

**Homework assignments**; these vary according to the class, but usually they include programming exercises that complement the material in the lecture notebook, and some non-coding tasks. The latter could be, *e.g.*, writing pseudocode for a given algorithm, commenting code line-by-line, or reflecting on how to approach a problematic issue such as a severe imbalance, or missing data. An example assignment for this unit is shown in Table 1.

In a traditional course, it is useful to have a final open-ended project that resembles a real-world application; importantly, this helps students create a "portfolio" item that can be added to a resume or brought up in job interviews. Students are encouraged to use Git and GitHub for their projects. In undergraduate settings, I have recently settled on a two-steps research problem. Step 1 sets a common goal for all students (*i.e.*, exploring and cleaning data, training and optimizing a model using one of the algorithms already discussed). In Step 2, I propose possible "tracks" that can be worked on by small group of students; students choose the "track". The "track" could consist in learning about and deploying a new algorithm, experimenting with feature engineering, analyzing the dependence on signal-to-noise ratio, and so on. In graduate-level classes, students are asked to come up with their own project and data, and the products include a project plan and a final report, ideally in the form of a research paper.

## 4. More about teaching to physicists

Many of the teaching tools highlighted in the previous sections are fairly general, and might be useful when teaching students from various backgrounds. So what aspects can we emphasize in particular when teaching to physicists? Here are some that come to mind:

- Literature search! This may sound like a given, but by asking students to read papers where these techniques are applied, and/or to search for related papers, we help them improve their literature searching skills, "skimming-through" skills, and of course we encourage building subject matter knowledge.

- Creating exercises that rely on domain knowledge is also important. For example, the last few exercises in Table 1 require an understanding of why photometric colors, which describe gradients in the spectrum, might be better features than measures of brightness, when the goal is to measure redshift. In the physical sciences, understanding the data before attempting feature engineering is very

important, and these applications can help reinforce why, as well as foster discussion of specific data issues.

- It is also useful to consider how machine learning approaches compare to other solutions. In the present example of photometric redshifts, it is possible to use templates of galaxy spectra to obtain an estimate of the redshift via inference methods. Physics and Astronomy students will be able to understand both approaches: a useful class discussion could be the comparison of pros and cons of inference versus machine learning.

- Finally, in Physics, we never want to quote a result without stating some level of confidence in our estimate. In this case, we could reflect on possible sources of uncertainties, from, e.g., the size of training set, the chosen architecture, and the available features, to the experimental uncertainties on the data; sometimes they are classified as *epistemic* and *aleatoric* uncertainties, see e.g. (Kendall & Gal, 2017). The former refer to the data set as a whole, while the latter can also be characterized on an object-by-object basis. Discussing how to estimate these uncertainties, and how to establish a fair comparison with existing results, is very important.

## 5. Conclusions

My conclusions are the following:

- It is important to include ML techniques in Physics (and more generally, STEM) curricula, as they are useful for both academic and non-academic careers;

- Using a mix of techniques, from traditional lectures to hands-on programming exercises, and recommending varied learning resources helps meet the needs of different learners;

- Practical projects are important, because they teach students to be comfortable with open-ended questions, and help them build a portfolio; students can be encouraged to post them on platforms like GitHub;

- Preparing good sets of materials is important not just for students, who tend to be resourceful and resilient, but also to widen the pool of instructors who can teach this subject;

- Persisting challenges include: 1. Improving access to data sets and problems at the right level of complexity and size; 2. Finding effective ways to teach across-discipline concepts, such as uncertainty estimation or interpretability, that are important but don't fit in the algorithm/problem mold, and 3. Integrating the conversation around ML topics with content taught *e.g.* in Statistics or Computational Methods courses.

All the materials from the most recent iteration of this class are available here (just send a sharing request, so I can keep track of who has access).

## Acknowledgements

A big big thank you goes to all my ML students throughout the years. You have inspired me so much, put up with me trying out new things, and provided great feedback, with kindness and humour. A special thanks to some of you from my first cohort: Hashir Qureshi, Harpreet Gaur, George Nwankwo, Kayla Ford, and Charlie Meyers, and to Ashwin Satyanarayana, who accepted to be my travel-mate in the crazy adventure of conjuring a course out of thin air and zero money as usual. I am also grateful to Andy Lawler, Elena Filatova, and Johann Thiel, who have kindly guest-lectured for my class at City Tech so many times, and have enriched the experience of students with their competence and passion. Here is to many more years of learning and improving!

Some of the work described here was partially sponsored by a CUNY "Research in the Classroom" Award (RIC 451). I am grateful to the Center for Computational Astrophysics of the Flatiron Institute for their continued hospitality and support.

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
