# OpenReview forum: "Teaching Machine Learning for the Physical Sciences:A summary of lessons learned and challenges"
_ecmlpkdd.org/ECMLPKDD/2021/Workshop/TeachML — TeachML 2021_

### Official Review · Reviewer_GAnr · 2021-07-04
**Specificities of teaching ML to physicists**

**Rating:** 8
**Confidence:** 4

**Review:**

The paper deals with the specificities of teaching ML for physical science, based on several years of experience by the author.

Example of such specificity is the difficulty to find dataset between the classic ML one (MNIST, Iris, etc..;) and the physics one used in Kaggle challenge requiring quite some background. The author then details his approach to teaching with a combination of formal session, reading material (including blogs and youtube videos), notebooks and final projects.
One interesting approach is to propose to the students notebooks which kind of work but with many suboptimal choices; students are invited to improve over all theses aspects.

The paper is quite interestingly detailing the author experience with teaching ML. I however regret that it does not quite fulfill the promises of dealing with the specificities of the physical science, like the control of uncertainties, the difficulty to find training data etc…

---

### Decision · Program_Chairs · 2021-07-23

**Decision:**

Accept

**Comment:**

Congratulations! Your paper has been accepted. The reviewer and the PCs agree that the paper is well written. The PCs do agree that this paper is less focused on the physical sciences than the title suggests.

Camera-ready version is due August 18, 2021. As you prepare the camera ready version, please take the reviewer's comments into consideration. The PCs recommend adding more specifics about teaching machine learning to physicists in particular.

We look forward to your participation at the workshop on September 13, 2021. We invite you also to join us for the satellite event on September 08, 2021. Schedules for both the workshop and the satellite event will be forthcoming.